# Scales++: Compute Efficient Evaluation Subset selection with Cognitive Scales Embeddings

## Abstract

The prohibitive cost of evaluating large language models (LLMs) on comprehensive benchmarks necessitates the creation of small yet representative data subsets (i.e., tiny benchmarks) that enable efficient assessment while retaining predictive fidelity. Current methods for this task operate under a model-centric paradigm, selecting benchmarking items based on the collective performance of existing models. Such approaches are limited by large upfront costs, an inability to immediately handle new benchmarks ('cold-start'), and the fragile assumption that future models will share the failure patterns of their predecessors. In this work, we challenge this paradigm and propose a item-centric approach to benchmark subset selection, arguing that selection should be based on the intrinsic properties of the task items themselves, rather than on model-specific failure patterns. We instantiate this item-centric efficient benchmarking approach via a novel method, Scales++, where data selection is based on the cognitive demands of the benchmark samples. Empirically, we show Scales++ reduces the upfront selection cost by over $18\times$ while achieving competitive predictive fidelity. On the Open LLM Leaderboard, using just a 1.0% data subset, we predict full benchmark scores with a 3.0% mean absolute error. We demonstrate that this item-centric approach enables more efficient model evaluation without significant fidelity degradation, while also providing better cold-start performance and more interpretable benchmarking.

## 1 Introduction

Large language models (LLMs) have demonstrated the ability to perform well on a broad range of tasks, including adapting to new tasks with little or no additional training (Brown et al., 2020). Evaluating LLMs across broad benchmark suites has become central to tracking progress, guiding training, and informing deployment (Raji et al., 2021). Yet running full evaluations is increasingly expensive in terms of energy and compute resources as models and datasets scale (Kaplan et al., 2020; Liang et al., 2023a; Hendrycks et al., 2021), and repeated re-evaluation during development exacerbates this cost.

To address this, recent work has focused on performing evaluation using small, carefully selected data subsets that can reliably predict a model's performance on the full dataset with high fidelity. Most existing approaches to selecting this subset of benchmark examples for scoring are **model-centric**: they construct the subset by exploiting similarities in *past model behavior*. For example, by clustering items using cross-model prediction statistics (Vivek et al., 2024) or by fitting psychometric (Item Response Theory - IRT) item parameters from historical item-level outcomes (Polo et al., 2024).

This assumption of access to item-level predictions of previous models on the target benchmark creates the following challenges: (i) It front-loads curation cost into running many models over many items (ii) It fails in *cold-start* regimes where comparable logs are unavailable (new/private model families) and (iii) It can struggle to *transfer* when behavior learned from one family does not generalize to another.

Table 1: Comparison of efficient LLM benchmarking methods. Model-centric approaches require extensive upfront cost due to the need for historical model evaluations, before evaluating new models. In contrast, our item-centric approach (SCALES++) achieves an $18\times$ reduction in setup cost while uniquely enabling cold-start evaluation through cognitive demand annotations rather than model performance patterns.

| Method | Paradigm | Core Assumption | Upfront Cost | No Historical Data Needed | Cold-Start Evaluation | Interpretable |
|---|---|---|---|---|---|---|
| Anchor Points (Vivek et al., 2024) | Model-centric | Past model correlations predict future correlations | $N$ models on full dataset ($N \geq 10$) | ✗ | ✗ | ✗ |
| tinyBenchmarks (Polo et al., 2024) | Model-centric | Past model failure patterns predict future patterns | 319 models on full dataset | ✗ | ✗ | ✗ |
| metaBench (Kipnis et al., 2025) | Model-centric | More past models $\rightarrow$ better future prediction | 5000+ models on full dataset | ✗ | ✗ | ✗ |
| **SCALES++ (Ours)** | **Item-centric** | **Cognitive demands of task items predict performance** | **16 annotations per item (1 with GNN)** | ✓ | ✓ | ✓ |

In this work, we challenge this dominant paradigm and propose a **item-centric** approach to benchmark subset selection. We argue that selection should be guided by the intrinsic properties of the task items themselves, rather than by model-specific failure patterns. We instantiate this approach with SCALES++, a novel method that captures the intrinsic cognitive demands of benchmark samples. Drawing inspiration from the General Scales framework (Zhou et al., 2025), which defines cognitive capabilities, we annotate each item along 16 cognitively grounded dimensions (e.g., logical reasoning, specific knowledge areas), yielding embeddings of item demands. We then (i) select a small, diverse subset in this space and (ii) predict full-benchmark performance via a combination of cluster-weighted estimates and per-dimension predictors, *without the need for any historical data*.

To amortize annotation costs across datasets, we distill General Scales using a lightweight Graph Neural Network (GNN) predictor trained on a small auxiliary dataset with ground-truth GPT-4o annotations. This predictor leverages frozen embeddings from a pre-trained, open-source LLM and requires only a single forward pass for scales prediction per benchmark instance, dramatically reducing upfront annotation costs. We term this approach SCALES++ LITE.

Our item-centric approach successfully addresses the limitations of prior work while maintaining competitive performance. Empirically, we demonstrate that SCALES++ reduces the upfront selection cost by over 18X while achieving high predictive fidelity. On the Open LLM Leaderboard, using just a 0.5% data subset, we predict full benchmark scores with a 2.9% mean absolute error; Our SCALES++ LITE annotates the entire in under 20 minutes, while outperforming expensive IRT baselines that require 16x more LLM calls by 0.2% MAE at 0.5% of evaluation data. We make three main contributions:

- We introduce a new item-centric paradigm for benchmark subset selection that overcomes the high costs and cold-start limitations of existing model-centric methods.

- We present SCALES++, a novel method that creates interpretable embeddings based on the cognitive demands of task items, moving beyond a reliance on model failure patterns. We also amortized annotations via our GNN predictor, allowing us to reduce per-item annotation costs for new datasets.

- We show on the Open LLM Leaderboard's six tasks that SCALES++ matches or surpasses model-centric baselines while cutting up-front costs by an order of magnitude.

For each benchmark in the Open LLM Leaderboard, we release our annotations as well as the selected subsets of the benchmark to be used for efficient benchmarking.

## 2 RELATED WORK

This work engages with works on efficient benchmarking and cognitive science in relation to LLMs.

## 2.1 EFFICIENT BENCHMARKING

The escalating computational costs of evaluating increasingly large language models have motivated substantial research into efficient benchmarking methodologies. Multiple studies have established that significant redundancy exists across benchmark items, with Ye et al. (2023) proposing to reduce the number of items in `Big-bench` (Srivastava et al., 2023), while Perlitz et al. (2024) demonstrated that evaluation on `HELM` (Liang et al., 2023b) relies on diversity across datasets but employs an excessive number of examples.

Building on these insights, benchmark curation and evaluation data selection methods have emerged as viable strategies for maintaining evaluation quality while reducing computational cost. Li et al. (2025) introduced the `BenchBuilder` pipeline, which leverages LLMs to curate high-quality prompts from large, crowd-sourced datasets through automated filtering based on seven quality indicators. Their approach was used to create `Arena-Hard-Auto`, a curated 500-item benchmark that capable of robustly recovering LLM relative rankings across multiple large benchmarks.

The closest to our work are recent efforts to perform evaluation using small, selected evaluation subsets that can predict a model's performance on the full benchmark. Vivek et al. (2024) proposed the Anchor Points method for evaluation subset selection, which advocates for reducing evaluation examples while maintaining accurate performance assessments. Polo et al. (2024) proposed the `tinyBenchmarks`, demonstrating that full performance can be reliably estimated on benchmarks such as `MMLU` and `HELM` within 2% mean absolute error leveraging trained IRT models (Item Response Theory) on evaluation results of 319 existing trained models on a small carefully curated subset of evaluation data. Most recently, Kipnis et al. (2025) introduced `metabench`, which compresses the entire `Open LLM Leaderboard` (Beeching et al., 2023)—a collection of LLM benchmarks—to less than 3% of its original size, providing reliable performance prediction and latent skill assessment by leveraging fitted IRT models trained on evaluation results from >5,000 trained LLM models. As highlighted in Table 1, a key challenge with these methods is the model-centric assumption that past model performance is helpful for selection. Consequently, these methods rely on historical data as the basis for selection, and hence have a significant upfront cost, before evaluating new models. We directly address this challenge with our item-centric SCALES++ approach which reduces the setup cost by 18x, while maintaining similar performance (see Sec. 3.2.2)

## 2.2 COGNITIVE APPROACHES

Recent work has begun exploring cognitive demand analysis as a means to better understand what LLM benchmarks actually measure by understanding the underlying cognitive requirements of evaluation tasks. This research direction seeks to decompose benchmark items into their constituent cognitive challenges, such as reasoning complexity, knowledge requirements, and processing demands, providing a more principled understanding of why certain tasks prove difficult for models.

The General Scales framework (Zhou et al., 2025) exemplifies this cognitive demand approach, operationalizing concepts from cognitive science to systematically analyze AI evaluation tasks. This framework operates by evaluating task items across multiple carefully crafted rubrics that systematically assess cognitive demands on scales ranging from 0 to 5, encompassing core cognitive abilities, knowledge domains, and task-related factors drawn from established cognitive science frameworks such as the Cattell-Horn-Carroll structure of human cognitive abilities (McGrew, 2005). The scales can be applied automatically using LLMs to annotate evaluation task items (see example in Appendix B), making the approach scalable to tag datasets.

While cognitive demand analysis was originally developed to understand and interpret benchmarks, we recognize its potential for addressing the orthogonal problem of efficient evaluation. Consequently, our work builds upon this foundation by adapting the General Scales framework for benchmark subset selection. We leverage the cognitive demand characterization provided by the 16-dimensional scale embeddings to identify representative evaluation instances, representing a novel item-centric approach to the problem of efficient benchmarking. By decomposing task items into their constituent cognitive demands, our method facilitates more principled selection of evaluation instances that preserve the cognitive diversity essential for comprehensive model assessment.

## 3 METHODS

### 3.1 PROBLEM SETTING

We consider the task of selecting a subset of items from a benchmark that best allows us to predict the overall score of a model on the benchmark. In this setting, evaluation of the model is costly and therefore only permitted on the selected subset of items, but the overall prediction may use other properties of the remaining benchmark items. This setting is similar to previous works (Polo et al., 2024; Kipnis et al., 2025), but we do not assume free access to the scores of other models on the benchmark.

More formally, we consider the task of predicting the performance of a model $\phi_m$ on a benchmark $B = (\{t_i\}_{i=1}^N, M)$ consisting of a set of items $\{t_i\}$ and a metric $M$ which assigns scores to each (model, item) pair, $M : \{\phi_m\} \times \{t_i\} \to [0, 1]$. Evaluating $M(\phi_m, t_i)$ is costly, so we want to select $I_{\text{sub}} \subset \{t_i\}_{i=1}^N$ with $|I_{\text{sub}}| = k \ll N$ such that we can predict $\sum_{i=1}^N M(\phi_m, t_i)$ without evaluating $\{M(\phi_m, t_i)\}_{t_i \notin I_{\text{sub}}}$. In this paper, we focus on the case where M gives a binary correct/incorrect score for each model generation, though in principle, the setup can be generalized to any metric.

We measure the cost of creating the overall predictions in terms of calls to an LLM, where, for simplicity, we count all LLMs as equivalent in cost. In our problem setting, evaluating a subset of the benchmark reduces the number of LLM calls from $N$ to $k$ per model, with additional upfront costs of $\ell N$ LLM calls to provide information about the benchmark items. The total cost of scoring $m$ models on the benchmark is then $\text{Cost}(m) = km + \ell N$, making $\ell$ a significant factor in determining the total cost of the evaluations in most cases.

### 3.2 ITEM SELECTION

#### 3.2.1 THE MODEL-CENTRIC SELECTION PARADIGM

Existing approaches operate under a **model-centric** paradigm. Such approaches assume access to historical, item-level behavior from a set of prior models, $\Phi = \{\phi_1, \phi_2, \ldots, \phi_n\}$, captured in a performance matrix $Y$. Prior methods like Anchor Points (Vivek et al., 2024) and `tinyBenchmarks` (Polo et al., 2024) use this matrix to guide subset selection. The process typically involves two stages: embedding and selection.

**1. Item Embedding** Each item $t_i$ is mapped to a low-dimensional embedding, $E_i$, that is a function of the collective performance of the source models.

- Direct Performance Embedding: The embedding for item $t_i$ is the vector of performance scores from all source models, i.e., the $i$-th column of $Y$. This is the basis for the Anchor Points method (Vivek et al., 2024), which uses the correlation between these vectors to define a distance metric for clustering.

- IRT-based Embedding: An Item Response Theory (IRT) model is fit to the entire performance matrix $Y$. The learned IRT parameters for item $t_i$ (e.g., discrimination $\alpha_i$ and difficulty $\beta_i$) form its embedding $E_i$. The `tinyBenchmarks` method uses this approach (Polo et al., 2024). The embedding $E_i$ is thus a function of the full matrix, $E_i = g_{\text{IRT}}(Y, i)$.

**2. Item Selection** A subset $I_{\text{sub}}$ is chosen by applying a clustering algorithm (e.g., K-Means or K-Medoids) to the set of all item embeddings $\{E_1, \ldots, E_N\}$.

Crucially, the entire selection process is a function of the performance matrix. i.e. $I_{\text{sub}} = f_{\text{mc}}(Y, k)$. This direct dependency on $Y$ leads to high upfront computation costs (to generate $Y$ by running $M$ source models on all $N$ items) and the inability to evaluate new models in cold-start scenarios where $Y$ is unavailable.

#### 3.2.2 THE ITEM-CENTRIC SELECTION PARADIGM (OURS)

To address these issues, we propose a **item-centric** paradigm that decouples the selection process from historical model performance. The selection function depends only on the intrinsic, observable properties of the task items themselves. Specifically, we assume a item-centric paradigm has an

*intrinsic* feature map $\psi : \mathcal{T} \to \mathbb{R}^D$ that depends only on the content of a task item $t_i \in \mathcal{T}$, not on any model's performance on it.

**1. Item Embedding** Each item $t_i$ is mapped to an embedding $C_i$ via a model-agnostic annotation function, $\psi$, that analyzes the content of the item. i.e. $C_i = \psi(t_i)$

In our work, SCALES++, we instantiate $\psi$ as a **Cognitive Scales annotation process** building upon Zhou et al. (2025). This function maps each task item $t_i$ to a 16-dimensional vector $C_i \in \mathbb{R}^{16}$, where each dimension corresponds to a specific cognitive skill or knowledge domain (e.g., logical reasoning, knowledge of social sciences). This annotation is performed using an LLM (e.g., GPT-4o) applied to a static, pre-defined rubric, making it independent of any specific model's success or failure on the item.

**2. Item Selection** As in the model-centric paradigm, a subset $I_{\text{sub}}$ is chosen by clustering the set of embeddings $\{C_1, \ldots, C_N\}$. In our implementation, we first use UMAP for dimensionality reduction before applying k-means clustering. The key distinction is that our selection process is a function of the task set $\mathcal{T}$, not the performance matrix $Y$. i.e. $I_{\text{sub}} = f_{\text{ic}}(\mathcal{T}, k)$.

By removing the dependency on $Y$, the item-centric paradigm inherently resolves the cold-start problem and dramatically reduces the upfront cost of subset selection. The annotation cost is also model-independent and can be amortized across all future model evaluations. In addition, in Sec. 3.3, we show how a distilled predictor can be used in place of the annotation function $\psi$, as an alternative model-agnostic annotation function.

BASELINE SUBSET SELECTION METHODS

**Random** We randomly select $k$ items from the benchmark, evaluate the target model on these items, and compute an average of the scores as a prediction of the overall score. For multi-task benchmarks such as the Open LLM Leaderboard, we use a weighted average such that each subtask is given equal weight in the overall score.

**Clustering** The scores from evaluating separate LLMs are used as an embedding for each item. Using these embeddings, the 'anchor points' are selected as the solution to a k-medoids problem (Rdusseeun & Kaufman, 1987). This approach is based on the Anchor Points method introduced in Vivek et al. (2024), though we use the more recent implementation of Polo et al. (2024), which selects the points closest to the k-means centers (McQueen, 1967) as the items to use for scoring. The overall score prediction is a weighted average of the score on these points, with weights proportional to the number of other points in the cluster corresponding to each anchor point.

**IRT** The scores from evaluating separate LLMs are used to fit an Item Response Theory (IRT) model to the benchmark items. We use the hyperparameters from (Polo et al., 2024), which introduced this approach, fitting a two-parameter 3-dimensional model. These parameters are then used as embeddings for the items in the benchmark, and points are selected with k-means clustering. The p-IRT estimator uses the learned IRT model with a weighted average of the scores on the cluster center items. For the gp-IRT estimator, this is combined with an estimator using a weighted average of the model scores on the cluster center items to form a final estimate. [1]

SCALES++ (OURS)

Our method begins by creating annotations for the degree to which each benchmark item requires 16 different cognitive skills (see Figure 1). These skills range from 'logical reasoning' to 'knowledge of social sciences', covering basic cognitive skills as well as knowledge in specific content areas. We use GPT-4o to annotate each item of the benchmark, using the rubrics developed and validated in Zhou et al. (2025). This creates a rating on a scale from 0-5 for each dimension.

We take these annotations as a 16-dimensional embedding of the benchmark items, which needs to be reduced to lower dimensions for effective clustering. We first discard any dimensions with no variation (This is possible if a benchmark has very similar items along a dimension. For example,

---

[1]Recent work in Kipnis et al. (2025) (metaBench) has noted an IRT model using results from only about 300 other models is likely to be underfit. They address this by training on results from >5000 other models and show better results, but due to the impracticality of having 5000 other model runs in most cases, we do not include their approach.

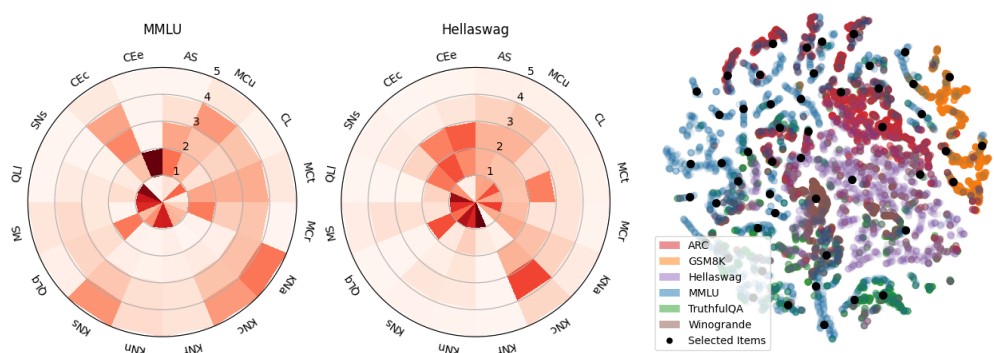

Figure 1: **Scales++ Item Selection.** For two example benchmarks, radial plots show the distribution of items which demand each level of capability along the 16 dimensions. MMLU often requires higher levels of subject area knowledge (KNa, KNc, KNs), but the two benchmarks have similar profiles along many dimensions, making the combined Open LLM Leaderboard a good candidate for subset selection. The embedded items are shown in a 2D t-SNE plot with the selected items highlighted in black, and the consituent benchmarks shown by the colors.

a mathematics benchmark is unlikely to require any social science knowledge in any item.) We then use UMAP (McInnes et al., 2018) to reduce the dimensionality of the embeddings from 16 dimensions to 3, and apply k-means clustering to select a subset of points. For the selected points, we evaluate the target LLM on each one, and estimate the overall benchmark score with a weighted average of the item scores weighted by the cluster sizes.

Leveraging the meaningful embedding dimensions, we fit a second estimator of item performance based on the difficulty levels of each item. We take the scores from the target model on each of the selected points from clustering and fit 16 separate logistic regressions for these points along each of the embedding dimensions. Based on the example of Zhou et al. (2025), we include additional data points with a performance of 0 at a difficulty of 20 representing a hypothetical maximum difficulty. For each remaining item in the benchmark, we predict the performance of the model using the average prediction of these regressions.

Our final estimator combines these two estimates with a weighted average. We use heuristic weights based on the results of Song & Schmeiser (1988) and Polo et al. (2024) for creating optimal linear combinations of estimators based on their bias and variance. Specifically, our final estimator uses weight $\lambda = \hat{b}_2^2/(\hat{b}_2^2 + \hat{v}_1)$ on the first, clustering-based estimator, and $(1 - \lambda)$ on the second logistic regression estimator, where $\hat{b}_2$ is the estimate of the bias of the logistic-regression estimator based on the selected items, and $\hat{v}_1$ is the estimate of the variance of the clustering-based estimator.

### 3.3 SCALABLE GNN-BASED PREDICTOR FOR COGNITIVE SCALES EMBEDDING

While our method demonstrates superior efficiency compared to prior approaches, the initial requirement of 16 GPT-4o calls per evaluation datapoint still incurs substantial computational costs. To further reduce this upfront expense, we train a lightweight neural network predictor that directly estimates the 16-dimensional cognitive scales embedding for any given task item, thereby eliminating the dependency on expensive GPT-4o inference.

The basic premise of our approach is to formulate the embedding prediction task as a standard supervised learning problem, where the objective is to replicate GPT-4o's cognitive assessment capabilities through a more computationally efficient model architecture[2]. To do this, we create a small set of auxiliary training data, which consists of 8,000 randomly subsampled queries from the Tulu3-SFT-mixture dataset (Lambert et al., 2025), and labelled it with ground-truth GPT-4o-generated cognitive scales embeddings. We train a lightweight classifier which leverages embeddings from a

---

[2]A natural approach would be to fine-tune a smaller LLM on GPT-4o outputs from identical query prompts used to obtain the scales embeddings. However, this approach still incurs significant computational cost from: (1) fine-tuning an LLM, and (2) autoregressive generation to label each point.

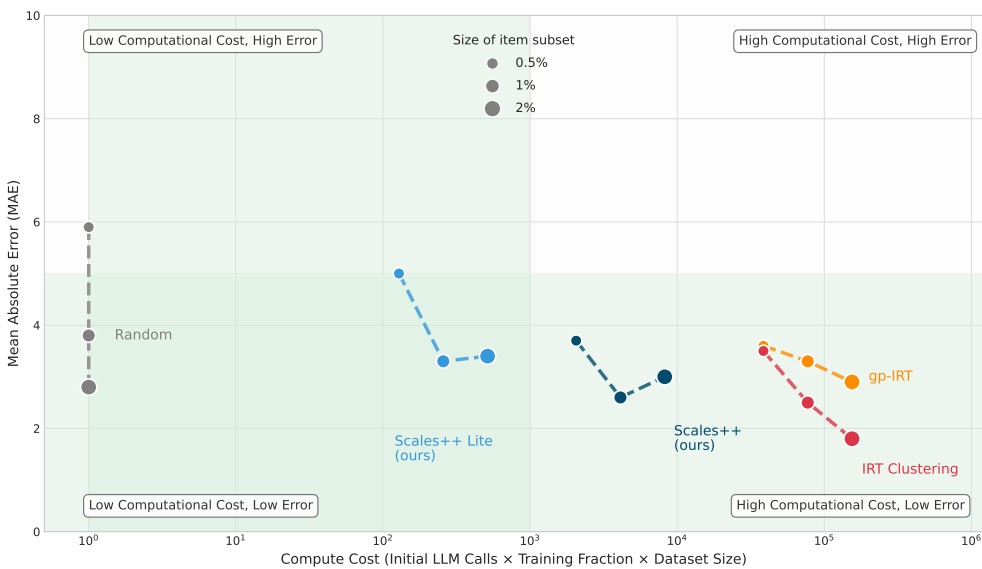

Figure 2: **Mean Absolute Estimation Error.** Performance estimation error in percentage points when selecting a subset of items to evaluate the target LLM and predict performance on the entire Open LLM Leaderboard. Marker size indicates the percentage of the benchmark being selected. The p-IRT is omitted due to being off the top of the chart, but is listed in Appendix A.

pre-trained LLM. Through empirical evaluation of various neural network architectures, we found that a Graph Neural Network (GNN)-based predictor consistently achieves the best performance on our validation dataset. To create the GNN, we embed each evaluation instance by feeding its query prompt to the Qwen2.5-7B-Instruct model (Qwen et al., 2025), from which we extract token embeddings at the 14th layer (middle layer) and apply mean pooling to obtain a fixed-dimensional representation for each sample. These LLM representations serve as node features in our graph construction, where we formulate the prediction task as node classification with 16-dimensional labels corresponding to cognitive scale dimensions, each ranging from 0 to 5. To construct the edges of the input graph, we connect each node and its top-10 nearest neighbours based on cosine similarity in the embedding space. The trainable GNN classifier comprises three stacked graph convolutional layers (Kipf & Welling, 2017) and is optimized using cross-entropy loss. Model selection is determined by validation performance on a held-out split of our auxiliary training data. This design offers significant computational advantages at both training and inference stages: LLM embeddings can be readily extracted from open-source models without expensive API calls, prediction for any task item requires only a single, non-autoregressive, forward pass through the LLM and classifier network, and the upfront training cost of this predictor can be amortized across multiple benchmark evaluations, making it increasingly cost-effective with additional evaluation samples.

Our GNN-based approach (i.e., SCALES++ LITE) significantly reduces the computational requirements for obtaining cognitive scale embeddings while maintaining prediction quality, enabling our benchmark subset selection method to scale to larger evaluation datasets without prohibitive costs.

## 4 RESULTS

We empirically evaluate the families of selection methods for efficient LLM benchmarking: (i) **model**-centric methods (clustering, IRT) and (ii) **item**-centric methods (ours), against a random baseline.

We test the performance of these estimation strategies on the Open LLM Leaderboard and six constituent benchmarks (Beeching et al., 2023). Combining test sets from GSM8K (Cobbe et al., 2021) (1319 items), MMLU (Hendrycks et al., 2021) (14042 items), Winograde (Sakaguchi et al., 2019) (1267 items), TruthfulQA (Lin et al., 2022) (817 items), Hellaswag (Zellers et al., 2019) (10042

| Subset | Method | MMLU | Hellaswag | TruthfulQA | GSM8K | Winogrande | ARC |
|--------|--------|------|-----------|------------|-------|------------|-----|
| 0.5% | Random | 3.9 (0.9) | 4.4 (3.2) | 17.4 (7.0) | 12.9 (2.7) | 11.7 (3.6) | 16.9 (8.4) |
| | IRT Clustering | 3.7 (0.2) | 3.3 (0.5) | 15.9 (3.2) | 14.8 (2.1) | 16.2 (3.0) | 18.5 (2.6) |
| | gp-IRT | 3.2 (0.5) | 2.5 (0.3) | 13.6 (2.5) | 14.3 (2.2) | 13.2 (2.2) | 14.0 (4.9) |
| | Scales++ | 4.7 (0.8) | 4.1 (1.2) | 12.2 (0.0) | 9.9 (1.1) | 12.7 (1.7) | 16.1 (5.4) |
| | Scales++ Lite | 4.9 (0.5) | 4.9 (2.2) | 18.5 (7.1) | 9.2 (1.0) | 16.4 (2.6) | 18.2 (6.9) |
| 1.0% | Random | 2.9 (1.0) | 3.0 (1.2) | 13.4 (9.4) | 8.5 (1.3) | 10.4 (0.3) | 13.0 (3.5) |
| | IRT Clustering | 2.7 (0.3) | 2.4 (0.3) | 12.0 (1.4) | 10.7 (0.7) | 9.9 (0.7) | 11.5 (1.5) |
| | gp-IRT | 2.5 (0.4) | 2.3 (0.4) | 10.3 (2.0) | 10.4 (0.8) | 8.0 (0.9) | 8.7 (2.2) |
| | Scales++ | 3.3 (0.5) | 2.9 (1.3) | 20.0 (3.6) | 9.1 (1.3) | 8.1 (2.5) | 12.1 (3.7) |
| | Scales++ Lite | 3.7 (0.8) | 2.5 (0.9) | 10.3 (3.2) | 9.1 (1.3) | 18.8 (0.1) | 10.7 (1.5) |
| 2.0% | Random | 2.1 (0.5) | 1.9 (0.6) | 10.2 (4.4) | 5.7 (0.6) | 6.8 (1.8) | 8.7 (3.9) |
| | IRT Clustering | 2.0 (0.1) | 1.7 (0.2) | 8.4 (1.6) | 8.3 (0.8) | 7.3 (1.0) | 8.5 (1.1) |
| | gp-IRT | 2.2 (0.4) | 2.1 (0.4) | 7.2 (1.8) | 8.1 (0.9) | 6.0 (1.3) | 6.6 (1.7) |
| | Scales++ | 2.5 (0.7) | 2.3 (0.8) | 12.8 (7.4) | 6.1 (1.2) | 6.0 (1.6) | 8.2 (2.8) |
| | Scales++ Lite | 2.4 (0.6) | 2.7 (0.8) | 5.9 (0.6) | 6.1 (1.2) | 8.4 (2.0) | 9.3 (3.2) |

Table 2: **Mean Absolute Estimation Error: Individual Benchmarks** Performance estimation error in percentage points for each benchmark. Values are reported as means over ten samples, with the standard deviation of the means.

items), and ARC (Clark et al., 2018) (1172 items), the benchmark contains 28,659 individual items, making it a prime candidate for down-sampling. More than 5000 models have been evaluated on the Open LLM Leaderboard with publicly released results, making it possible to collect item-level scores for testing for free. For comparison with previous works, we use scores from the same subset of 395 models as were used for evaluation in (Polo et al., 2024).

We compare the evaluation results for three different sizes of the benchmark subset, from 0.5% to 2.0% of the total benchmark. For each method, we collect ten repetitions with different random seeds to show the effect of the non-determinism present in k-means, IRT, and UMAP. Unless otherwise stated, from the 395 models collected from the Open LLM Leaderboard, we hold out the scores of the 95 most recently-released models as the test set and report mean absolute error (**MAE**, ↓ better) between predicted evaluation scores on the subset vs evaluation on the full dataset.

Figure 2 shows a comparison of SCALES++ and SCALES++ LITE to baseline methods, both indicating compute cost and error prediction. We find that SCALES++ achieves 2.6% MAE when sampling just 1.0% of the benchmark (286 items), outperforming random selection by 32% and matching IRT-Clustering's performance while requiring 95% fewer initial LLM calls. We observe particularly strong results for SCALES++ LITE, reducing compute cost by another order of magnitude relative to SCALES++ with only a limited decrease in predictive power. Remarkably, SCALES++ LITE can annotate the entire Open LLM Leaderboard, including 28,659 evaluation instances, in under 20 minutes while outperforming or maintaining competitive performance against IRT baselines under evaluation data scarcity (e.g., 1.0%).

## 4.1 INDIVIDUAL BENCHMARK RESULTS

We conduct the same testing for each of the constituent benchmarks of the Open LLM Leaderboard. For each benchmark, we select subsets of 0.5%, 1.0%, and 2.0% and report the mean MAE on the held out models.

Table 2 shows the results for each of the methods and benchmarks. gp-IRT is typically the strongest method, but Scales++ within 2% MAE (or better) in 70% of cases, while requiring 95% fewer LLM calls to create. TruthfulQA is particularly challenging for the Scales++ method, but this appears to be a result of the embedding annotations rather than the item-centric framework, as Scales++ Lite is the best approach for 1.0% and 2.0% samples. GSM8K, where Scales++ consistently outperforms

the IRT methods, has very dense Scales embeddings, with only 52.3% of benchmark items having a unique embeddings, compared to $\geq 84\%$ for the other benchmarks, and as high as 96.6% for TruthfulQA.

# 5 EXPERIMENTS

Below we analyze the capabilities of these benchmark subset selection methods to generalize across architectures and model size. We provide additional experiments and ablations in Appendix A

## 5.1 CROSS-ARCHITECTURE GENERALIZATION

We explore how well item selection strategies learned from one model architecture effectively transfer to another. We use this as a proxy for the problem of generalizing to new model architectures. We conduct two sets of transfer experiments, between Dense and MoE models, and from Llama-based models to all other architectures. We are limited in our ability to compare architectures which differ more significantly due to the available data.

For each experiment, we predict the performance of models in one category using only training data from the other category. (This is only relevant for the baselines, since SCALES++ does not use previous model runs for training data.) MoE models are relatively rare in the dataset, so for MoE→Dense we train on results from 16 randomly selected MoE models and test on all 363 dense models, and for Dense→MoE we train on results from 16 randomly selected dense models and test on all 32 MoE models.

Table 3: Cross-architecture generalization: MoE → Dense

| Framework | Method | 0.5% | 1% | 2% |
|---|---|---|---|---|
| | Random | 5.8 ± 0.5 | 4.5 ± 2.2 | 2.7 ± 2.2 |
| IRT | Clustering | 2.4 ± 1.9 | 2.2 ± 1.3 | 2.6 ± 1.6 |
| | gp-IRT | 3.3 ± 2.3 | 2.2 ± 1.5 | 2.6 ± 1.6 |
| Scales | SCALES++ | 3.7 ± 2.8 | 2.2 ± 1.7 | 1.8 ± 1.1 |

Table 4: Cross-architecture generalization: Dense → MoE

| Framework | Method | 0.5% | 1% | 2% |
|---|---|---|---|---|
| | Random | 7.4 ± 5.1 | 3.7 ± 2.5 | 2.6 ± 2.9 |
| IRT | Clustering | 3.9 ± 3.1 | 2.7 ± 1.6 | 1.8 ± 1.2 |
| | gp-IRT | 4.0 ± 3.5 | 3.9 ± 2.6 | 2.6 ± 2.2 |
| Scales | SCALES++ | 4.3 ± 3.8 | 2.3 ± 1.5 | 2.2 ± 1.5 |

For Llama-based models, we train on results from 205 Llama-based models for training, and test on the results from the remaining 123 labelled models, excluding 67 models with no architecture label. For each experiment, we conduct ten repetitions with different random seeds and report the mean performance.

Table 5: Cross-architecture generalization: Llama models → All others

| Framework | Method | 0.5% | 1% | 2% |
|---|---|---|---|---|
| | Random | 5.5 ± 2.2 | 3.9 ± 2.0 | 2.8 ± 1.4 |
| IRT | Clustering | 3.3 ± 0.3 | 2.3 ± 0.3 | 1.9 ± 0.3 |
| | gp-IRT | 3.7 ± 0.7 | 2.8 ± 0.5 | 2.8 ± 1.0 |
| Scales | SCALES++ | 3.9 ± 1.3 | 2.6 ± 0.9 | 2.8 ± 0.4 |

Tables 3, 4, and 5 show successful generalization across architectures with $< 2\%$ MAE for sufficiently large item subsets. Both frameworks show better than random cross-architecture generalization.

## 6 DISCUSSION & CONCLUSION

As the rapid development of LLMs continues, the need for efficient and reliable evaluation methods becomes increasingly critical. This work introduces a shift in approach from model-centric to item-centric benchmark subset selection, addressing fundamental limitations in current efficient benchmarking approaches. SCALES++ provides the best practical efficiency-accuracy trade-off, achieving <3% error with only 1.0% of items and minimal initialization cost. The comparison between 16 vs 300 LLM calls for initialization reveals IRT's hidden computational cost, making our approach 18X more efficient for comparable accuracy. This makes SCALES++ particularly valuable when evaluating on new benchmarks or working under computational constraints, as it translates to concrete benefits: *a 70B parameter model can be benchmarked in hours rather than days*. Furthermore, the SCALES++ LITE variant democratizes efficient benchmarking by reducing annotation costs through our GNN-based predictor, enabling comprehensive benchmark annotation in under 20 minutes while maintaining competitive performance in predictive accuracy.

We note a few limitations relevant to the application of our method. The performance is generally competitive with the existing methods, but the main benefits come from the reduced cost. In situations where sufficient previous data is already available, all of the methods should be considered. Our testing is also limited to the Open LLM Leaderboard, which has sufficient previous data for the baselines. This setting also has a variety of task items, which means the Scales embeddings will have greater dimensional variation. In single domain settings, the embeddings may be less useful. On the individual benchmarks, we found mixed results relative to the baselines, with possible indications that embedding density could be explored as an indicator of domains where Scales++ is more likely to be effective. We also note that all of the methods tested show relatively high variance between random seeds, an issue which could potentially be addressed in the future by combining these methods with adaptive testing methods such as in the recent work of Hofmann et al. (2025).

Our work demonstrates that focusing on intrinsic task properties rather than historical model behavior offers a more efficient and potentially more generalizable path forward for comprehensive LLM assessment. The ability to achieve competitive mean absolute prediction error without training on the results of previous models represents not just an incremental improvement, but a fundamental rethinking of how we approach the increasingly important challenge of LLM evaluation at scale.

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

# A  ABLATIONS AND ADDITIONAL EXPERIMENTS

## A.1  CLUSTERING METHODS

Clustering algorithms are used in combination with the task representations to select the final subset of tasks. The goal of this step is to select the points which best represent the benchmark as a whole. If the ability of a task to represent another task is measured by distance in the embedding space, then this problem is equivalent to a k-medoids problem, finding the set of $k$ points which minimizes the average distances to all of the other points in the dataset. For efficiency, we approximate the solution with the task item closest to the k-means centers, potentially affecting downstream performance.

In this appendix, we validate the effectiveness of k-means across frameworks or if alternative clustering methods offer advantages. We compare K-MEANS, K-MEDOIDS, and GMM for selecting a subset of points with both IRT- and Scale-based embeddings. Since we are focusing on the clustering methods, we do not report results for the random baseline. We report the mean MAE across the $0.5\%, 1.0\%$ and $2.0\%$ subsets.

Table 6: IRT - Clustering method comparison (mean MAE).

| Clustering | CLUSTERING (Avg) | GP-IRT (Avg) |
|---|---|---|
| K-MEANS | **2.30** | **2.30** |
| K-MEDOIDS | 2.50 | 2.56 |
| GMM | 2.81 | 2.86 |

Table 7: Scales - Clustering method comparison (mean MAE).

| Clustering | CLUSTERING (Avg) | SCALES++ (Avg) |
|---|---|---|
| K-MEANS | **2.48** | **2.41** |
| K-MEDOIDS | 3.66 | 3.54 |
| GMM | 3.30 | 3.26 |

Tables 6 and 7 show that K-means consistently achieves the lowest MAE. K-medoids and GMM show 10-160% higher error rates, particularly for Scales-based methods.

## A.2  NUMERICAL RESULTS FOR FIGURE 2

| | Initial | *Open LLM Leaderboard* | | |
|---|---|---|---|---|
| | LLM calls | 0.5% | 1.0% | 2.0% |
| Random | 0 | 5.9 (2.3) | 3.8 (1.7) | 2.8 (1.3) |
| IRT Clustering | 300 | 3.5 (0.4) | 2.5 (0.3) | 1.8 (0.2) |
| p-IRT | 300 | 10.7 (3.3) | 12.0 (4.0) | 12.1 (3.9) |
| gp-IRT | 300 | 3.6 (0.6) | 3.3 (1.3) | 2.9 (1.3) |
| Scales++ (ours) | 16 | 3.7 (1.2) | 2.6 (0.8) | 3.0 (0.9) |
| Scales++ Lite (ours) | 1 | 5.0 (1.8) | 3.3 (0.7) | 3.4 (1.1) |

Table 8: **Mean Absolute Estimation Error:** Performance estimation error in percentage points when selecting a subset of *n* items to evaluate the target LLM and predict performance on the entire benchmark. Small numbers are the sample standard deviation across 10 repetitions. The Open LLM Leaderboard contains 6 sub-benchmarks, and we allow the subset to be selected from any of the question items. We include p-IRT here but not in the main figure due to scale.

## B  GENERAL SCALES

General Scales (Zhou et al., 2025) represents a comprehensive framework for AI evaluation that can explain what common AI benchmarks really measure, extract ability profiles of AI systems, and predict their performance for new task instances. The methodology builds on 18 newly-crafted rubrics that place instance demands on general scales that do not saturate, providing a standardized approach to assess cognitive and knowledge-based abilities across diverse AI evaluation tasks. The list of dimensions used in our works include:

- Attention and scan
- Calibrating knowns and unknowns
- Conceptualisation learning abstraction
- Critical thinking processes
- Identifying relevant information
- Knowledge applied science
- Knowledge customary
- Knowledge formal science
- Knowledge natural science
- Knowledge social science
- Logical reasoning
- Mind modelling and social cognition
- Quantitative reasoning
- Spatial reasoning and navigation
- Verbal comprehension
- Verbal expression

These scales are obtained through an automatic annotation process using GPT-4o, with each task instance rated from 0 to 5 on each dimension based on detailed rubrics, indicating how much that ability contributes to successful task completion.

Below is one dimension-specific prompt template, where {{instance}} is replaced with the prompt from the task instance in the evaluation benchmark.

> **Prompt for Attention and Scan**
>
> QUERY: The following rubric describes six distinct levels of *Attention and Scan* required by different tasks
>
> # Attention and Scan (AS)
>
> This criterion assesses the level of attention and scan required to focus on or locate specific elements within a given stream of information or environment in the whole process of solving a task. During this process, there is the need to actively scan for or retrieve elements that meet predetermined criteria. The level represents the extent to which the task requires locating and focusing on specific target information, ranging from situations where the target is immediately obvious to those requiring sustained tracking of multiple targets among numerous distractors—any elements that are irrelevant to solve the task, such as visual objects, sounds, pieces of text, noise, or other stimuli, but compete for attention with the target information—in complex, dynamic environments. The challenge is not on determining what to look for but focusing the attention to find it within a larger context. This differs from tasks where there's a need to identify which pieces of information are relevant from a set already under consideration. While both processes may overlap in complex tasks like reading comprehension or image understanding, "attention and scan" specifically focuses on the deployment of attention during scan processes when solving the task, rather

than the selection or evaluation of information.

## Levels

### Level 0: None
No attention or scan is required. The target information is immediately obvious or is the only information present.
**Examples:**
- "Given a single word input, determine if it starts with a capital letter."
- "Look at the only object in the centre of the white page and tell what colour it is."
- "Is Madrid the capital of Spain?"

### Level 1: Very low
Minimal attention or scanning is required. The target information is easily distinguishable with little to almost no distraction.
**Examples:**
- "Find the only blue car in a car park full of red cars."
- "Find the letter 'X' among a row of 'O's"
- "Spot the tall tree in a row of short bushes."

### Level 2: Low
Some attention or basic scanning is required. The target information is visible among a few distractors or in a small scan area.
**Examples:**
- "Find all the vowels in the following sentence: 'The quick brown fox jumps over the lazy dog.'"
- "Find who's wearing glasses in this photo of students at commencement, with 2 rows of 5 students each, all facing forward, taken by a professional photographer."
- "Who authored the Queensberry rules, which were published in 1867 for the sport of boxing? Choices: A. John Douglas (in his late twenties) B. John Graham Chambers (in his mid-twenties) C. Marquess of Queensberry (in his early thirties) D. James Figg (in his forties)."

### Level 3: Intermediate
Moderate attention and scan are required. The target information is mixed with several distractors or spread over a fairly large scan area.
**Examples:**
- "Find everyone wearing glasses in this casual BBQ photo where 15 people are gathered around a table. Some are sitting, some standing, some looking at the camera while others are in conversation."
- "In a 5-page technical document about basic geometry, locate all explicit references to the Pythagorean theorem ($a^2 + b^2 = c^2$), where the equation appears 5 times mixed among references to 15 other geometric formulas, with occasional inconsistent equation numbering but standard mathematical notation."
- "As we all know, the Queensberry Rules are a set of rules for boxing that govern both amateur and professional matches. Who authored the Queensberry rules, which were published in 1867 for the sport of boxing? Choices: A. John Douglas (in his late twenties) B. John Graham Chambers (in his mid-twenties) C. Marquess of Queensberry (in his early thirties) D. James Figg (in his forties) E. James Zou (in his fifties) F. Lucy Grande (in her late twenties) G. Xiaoxiao Li (in her early forties) H. Enrique Garcia (in his late thirties)."

### Level 4: High
Sustained tracking of one or various targets is required. The target information is in an environment mixed with numerous distractors and changing conditions. Requires some continuous monitoring amid competing signals.
**Examples:**
- "Listening to a symphony, identify all instances where the clarinet plays in a minor key,

even when it's not playing the main melody."
- "Track three orange spheres among twenty red spheres as they move randomly across a black screen (40 cm × 30 cm) at varying speeds (1-3 cm/s), with spheres frequently intersecting paths and maintaining a minimum separation distance of 2 cm. Each sphere is 1 cm in diameter."
- "In a real-time video feed of a busy airport, finding the locations of ten blue suitcases."

### Level 5+: Very High
Requires sustained attention and scan for simultaneous tracking of multiple targets across different domains or contexts, with continuous adaptation to fast-changing conditions. The target information is extremely difficult to distinguish from distractors or is hidden in a vast or constantly changing environment.
**Examples:**
- "While seated courtside at a professional basketball game, track two specific players throughout the entire game as they move at speeds up to 8m/s, frequently cluster with other players during rebounds, and weave through screens and defensive formations."
- "Monitor four simultaneous video feeds of a crowded airport terminal from different angles, detecting subtle security-relevant changes (e.g. brief interactions ¡ 2 seconds, crowd flow changes, small object exchanges) across feeds."
- "While monitoring multiple simultaneous customer service chat conversations in different languages, identify instances where customers are expressing the same underlying technical issue, even though they're describing it using different metaphors, technical terms, or cultural references specific to their region."

TASK INSTANCE: {{instance}}

INSTRUCTION: Score the level of *Attention and Scan* demanded by the given TASK INSTANCE using a discrete value from 0 to 5. Use CHAIN-OF-THOUGHTS REASONING to reason step by step before assigning the score. After the CHAIN-OF-THOUGHTS REASONING STEPS, conclude your assessment with the statement: "Thus, the level of *Attention and Scan* demanded by the given TASK INSTANCE is: SCORE", where 'SCORE' is the integer score you have determined.

