# OpenReview forum: "Scales++: Compute Efficient Evaluation Subset selection with Cognitive Scales Embeddings"
_ICLR.cc/2026/Conference — Submitted to ICLR 2026_

### Official Review · Reviewer_4sK9 · 2025-10-24

**Soundness:** 2
**Presentation:** 1
**Contribution:** 3
**Rating:** 2
**Confidence:** 3

**Summary:**

This paper proposes a new method, Scales++, to predict benchmark performance, based on evaluating LLMs on a limited number of items. The method uses GPT-4 to embedd benchmark items in a low-dimensional space, reflecting different cognitive capabilities necessary to solve the item. Items are then clustered in this space, and evaluation focuses on a number of items that are representative for each cluster. The authors claim that their method predicts benchmark performance similarly well as existing methods, while not requiring historical evaluation data of other LLMs.

**Strengths:**

- Using item characteristics to predict benchmark performance is a promising direction
- The empirical performance of the proposed method appears to be strong

**Weaknesses:**

- It is quite difficult to understand the experimental setup for most of the experiments in section 5, making it difficult to evaluate (or reproduce) these experiments.
- I suspect that the evaluation of the random baseline in figure 2 is bugged: MAE should approximately decrease as 1/sqrt(n) in the sample size n, but essentially stays constant going from 0.5% to 2% of items. It is unclear, whether the cause of this could affect the evaluation of the more complicated alternatives as well.
- There is no meaningful discussion of limitations. Given that the method does not come with any guarantees (and seems to severely underperform alternatives in some settings, according to the appendix), disclaimers about how to responsibly use this should be included in the main text.
    - In light of the brittleness of the proposed frameworks performance gains, the main text's focus on one large test set, aggregating multiple benchmarks, also feels a bit misleading.
    - It also seems worth noting, that the computational gains of the proposed method only manifest for new benchmarks, as comprehensive evaluations of a large number of models are publically available for most existing benchmarks.
- Beyond that, the paper would strongly benefit from a thorough proofread. Examples:
   - Table 8 seems to be missing entries for IRT++
   - The writing refers to a 10% subset setting multiple times, but the results only seem to report up to 2%
   - "Our SCALES++ LITE annotates the entire in under 20 minutes" missing word
    - "Their approach successfully created Arena-Hard-Auto, a curated 500-item benchmark that capable of robustly recovering LLM relative rankings across multiple large benchmarks" missing word, also "successfully created" is slightly strange phrasing
    - "This work introduces a paradigm shift" seems overly grandiose
   - "Our work demonstrates that focusing on intrinsic task properties rather than historical model behav-ior offers a more robust and efficient path forward for comprehensive LLM assessment." The robustness claim does not seem to be substantiated anywhere in the paper

**Questions:**

- What is the setting with 16 initial LLM calls for IRT reported in Table 2? Also, why are these results not reported in Figure 2 (IRT seems to outperform the proposed method for >0.5% data points here)?

---

> ### Author Response · Authors · 2025-11-26
>
> Thank you for your review. We appreciate your thoroughness and belief in the value of the contribution. We have made substantial changes based on your feedback which have strengthened the paper.
>
> At a high level the changes are:
>
> 	1. **Bugfix on the random baseline.** As two reviewers noticed, there had previously been a bug where the random baseline used too few points in the sample. The random baseline now uses the same number of points as the other methods and is stronger.
> 	2. **Text clarification and proofreading.** We apologize for the proofreading errors in the first version. We have now edited more closely.
> 	3. **New aggregation approach.** In response to comments from the first reviewer, rather than treating every item as equal, we are now weighting the items so that each subtask of the leaderboard is given equal weight. We have re-run all of the results with this change. We highlight this to explain why the numbers have all changed. (We also re-labelled IRT++ to gp-IRT to avoid confusion, and added p-IRT in the appendix.)
>
> Overall, we still find that our method achieves results comparable to the top baselines with far less upfront cost.
>
> On the specific points:
>
> - We have re-written Section 5 to clarify what we are doing in the experiments and to make them better reproducible.
> - We have corrected the random baseline. It was not correctly changing the number of points being sampled, as you suspected. Thank you for pointing this out with a clear intuition
> - We have added a limitations section. We have also included indications of when the method might be expected to perform better or worse, though the key use case is still for benchmarks where the typical approaches are not possible due to lack of previous data.
> - We have added the results on the individual benchmarks within the Open LLM Leaderboard. These results help to show the performance of the method across different benchmarks as well as giving a sense of the cases where it is likely to perform better or worse.
> - We did not run sizes above 2%, and have fixed the text.
> - We have tried to soften the language around “paradigm shift”. We do think “paradigm” is an accurate word, but agree the connotation can be grandiose. We are open to other suggestions.
> - The experiments in Section 5 are intended to test the robustness of the various methods when generalizing across different model architectures. Our approach does well here, but is not clearly better than the other methods, so we have changed the language to focus on efficiency. In principle, we do believe it is a weaker assumption to assume that the cognitive demands of problems generalize across models than to assume that patterns of strengths and weaknesses in present models are predictive of future ones.
>
> Thank you again for your helpful review. We believe it has improved our paper!

---

> > ### Comment · Reviewer_4sK9 · 2025-11-27
> >
> > Thank you for updating the paper and fixing the baselines.
> >
> > Overall, I remain a bit confused about the intented use-case for the proposed method: One one hand, it seems to perform worse than previously proposed methods when not taking into account compute costs. But on the other hand, compared to the random sampling baseline, it is substantially more expensive in terms of compute, does not come with formal performance guarantees, and performs worse in 9/18 of the experiment conditions considered in Table 2.

---

### Official Review · Reviewer_pZpu · 2025-10-31

**Soundness:** 3
**Presentation:** 3
**Contribution:** 3
**Rating:** 6
**Confidence:** 4

**Summary:**

The paper presents a new perspective to efficient evaluation of language models - using item-centric view instead of the model-centric view.
In this view, the proposed method first rate each evaluation example using 16 criteria (first introduced in a prior work (Zhou et al. 2025)), and use these scores as the representation ("embedding") of the example. This is more efficient than relying on performance of prior models as the representation, which could involve 300+ models.
The paper further present a more efficient variant of the method, that further reduce the 16 LM calls for the 16 criteria, and uses a GNN trained with auxiliary data instead.
Empirically the method is comparable with model-centric approaches while being more computationally efficient.

**Strengths:**

* Taking on an example-centric view is under-explored for this problem and this work innovatively studies this. The method is also shown to be effective.
* The paper also did a great job in introducing prior works and contrasting them with the proposed method.

**Weaknesses:**

* Several remaining confusions about methods and experiment settings. See questions below.

**Questions:**

* Can you elaborate more on some design choices you made? Why is the UMAP dimension reduction needed? Why is Graph neural network preferred over other possible networks for classification?
* Can you elaborate more on the auxiliary training data you used? It was currently first mentioned in Line 316 without much introduction. In line 343 you further mention a validation split of auxiliary data. Is the full TULU-SFT set used as auxiliary data? How large is that?
* Can you explain in figure 2, what are included in compute cost for each method? Is "initial LLM calls" equal to 16 for scales++ and 300 for IRT methods?
* I'm quite confused at section 5 where some cross-architecture and cross-model-size experiments were done. In my understanding scales does not require any training on "source models", so what does "cross-architecture" refer to here?

Minor:
* Line 113: The citation of BIG-bench needs to be corrected.

---

> ### Author Response · Authors · 2025-11-26
>
> We thank the reviewer for their positive and insightful feedback. We are delighted that they found our "example-centric view" to be "under-explored" and "innovative," and that they appreciated the paper's presentation and comparison to prior work. We are happy to answer your questions about implementation details, and have made additional revisions to the manuscripts to provide greater clarity as well.
>
> 1. On Design Choices (UMAP and GNN)
> *Why is UMAP dimensionality reduction needed?*
> Our clustering problem is theoretically similar to k-medoids, where we want to select the subset of points within the dataset which minimize the average distance from a point in the dataset to one of the subset points. In practice, we use k-means to approximate k-medoids. Our cognitive scales annotation process results in a 16-dimensional embedding for each task item. While 16 dimensions is not high-dimensional in all contexts, clustering algorithms like k-means can still be affected by the curse of dimensionality, where distances between points become less meaningful.
>
> Consequently, we chose UMAP to reduce to 3D for two reasons: (1) Improved clustering behaviour and (2) Consistency with IRT baselines like tinyBenchmarks which uses a 3-dimensional IRT embedding.
>
> *Why is a Graph Neural Network (GNN) preferred?*
>
> For our SCALES++ LITE variant, we needed a predictor to estimate the 16 cognitive scale dimensions. While a standard classifier like an MLP could in theory, be used, we opted for a GNN to amortize the annotation cost, given the capability of a GNN to capture structural dependencies between samples within the LLM embedding space. Specifically, we embed each of the benchmark items with Qwen2.5-7B-Instruct and construct a graph where each sample (benchmark item) is a node with edges connecting each node to its 10-nearest-neighbors based on cosine similarity in the embedding space. This graph structure allows the GNN to aggregate information from similar-looking items when making its prediction for a given item. i.e. Benchmark items with similar cognitive demands should benefit from information sharing,  leading to a more robust and accurate estimation of the cognitive scales.
>
> 2. On the Auxiliary Training Data
> We apologize for the lack of clarity on the auxiliary dataset. To train our GNN predictor, we first needed a labeled dataset.
>
> Source Data: We used the Tulu3-SFT-mixture dataset as a large, diverse source of prompts.
> Sampling: We randomly subsampled 8,000 samples.
> Annotation: We then generated our ground-truth labels by annotating these 8,000 queries using the full GPT-4o-based 16-scale annotation process.
> Training: This created a labeled dataset of 8,000 (prompt, 16-D cognitive vector) pairs. We then use a 5:1 train/valid split on this 8,000-sample set to train and select the best-performing GNN predictor.
> (**This 8000 X 16 upfront cost was included in Figure 2 for Scale ++ LITE)
> **We have updated Section 3.3  to be clearer about this process.**

---

> > ### Author Response · Authors · 2025-11-26
> >
> > 3. On the Compute Cost in Figure 2
> > Your understanding is correct. The "initial LLM calls" ($l$) is the number of LLM calls per item needed to generate the embeddings for selection. As shown in Table 8, this is:
> > IRT Methods: 300 (one call for each of the 300+ models).
> > SCALES++: 16 (one call for each of the 16 cognitive dimensions).
> > SCALES++ LITE: 1 (a single forward pass through our GNN).
> >
> > The x-axis in Figure 2 plots the total upfront cost ($l \times N$), which is this $l$ value multiplied by the total number of items in the benchmark ($N=28,659$). This is the cost incurred before any new model can be evaluated.
> >
> > We scale this upfront cost ($l \times N$) by the fraction of the dataset which is selected ($f$). We do this as an approximation of the cost of running evaluations for $k$ models, which would be ($l \times N + f \times N \times k$) without the dependence on $k$, which is representative when $k$ is not too large. We plot this on a log scale to visually emphasize the orders-of-magnitude cost difference between the high computational cost model-centric paradigm and our low computational cost item-centric one.
> > **Note:** This deemphasizes the per-model evaluation cost ($f \times N \times k$), which is the same for any method. We are focused on upfront selection costs for fixed $f$.
> >
> > 4. On the Cross-Architecture Experiments (Section 5)
> > This is a critical point of clarification, and we thank you for asking. The reviewer is correct that SCALES++ does not require any "training" on source models. This is its main advantage.
> > The experiment in Section 5.1 is designed to test the robustness of the subsets selected by each paradigm when applied to model architectures unseen during the selection process.
> > -  	For Model-Centric (IRT) Methods: "Training" refers to fitting the IRT model on the performance data from a specific set of models (e.g., the 205 Llama-based models). This "learns" a subset based on Llama failure patterns. We then test how well this Llama-derived subset predicts performance for non-Llama models.
> > -  	For Our Item-Centric (SCALES++) Method: There is no "training."  We simply evaluate based on our cognitive selection on different slices of the model set to show that their MAE remains competitive across architectures and model sizes.
> >
> > The goal is to show that our model-agnostic subset generalizes to new architectures just as robustly as the model-centric subsets that were explicitly trained on a specific architecture. This highlights that even though we don't "train" on source models, we validate that our cognitive-based predictions are not only cheaper but work equally well for Dense, MoE, and different architecture families
> > **We have added an explicit clarification in Section 5.1 for this point.**
> >
> > 5. Minor Point
> > Thank you for catching the citation error for BIG-bench. We have corrected this in the updated manuscript.
> >
> > We appreciate the reviewer's constructive feedback, which will help us significantly improve the paper's clarity.

---

> > > ### Comment · Reviewer_pZpu · 2025-11-28
> > >
> > > Thank you for clarifying the design choices and auxiliary data usage. Those were very helpful for understanding your experiments. I still think an item-centric view is underexplored and this paper took some meaningful steps.
> > >
> > > I read the other two reviews and responses. I noticed significant changes to the main result figure (figure 2) during the rebuttal. The updated results seems very mixed now, e.g., (1) performance of Scale++ unexpectedly decrease when the size of the subset increase (2) performance of Scale++ do not consistently outperform the random baseline. This is quite concerning and I want to ask (1) how do you interpret this set of results? what might be causing the two observations above? (2) when should someone choose Scale++ or Scale++ Lite over a random baseline?

---

### Official Review · Reviewer_Gonq · 2025-10-31

**Soundness:** 1
**Presentation:** 3
**Contribution:** 1
**Rating:** 2
**Confidence:** 4

**Summary:**

The paper proposes an item-centric approach for benchmarking subset selection to enable efficient evaluation. The authors argue that previous methods suffer from high upfront costs, cold-start issues, and potential generalization problems. To address these limitations, they cluster data points using the "General Scale," which leverages GPT-4o to annotate 16 cognitively grounded properties per data point. To reduce cost further, they train a GNN to predict these 16 dimensions. Experimental results demonstrate the method’s efficiency.

**Strengths:**

- The paper is well presented and easy to follow.
- It highlights important issues with current efficient evaluation techniques, some of which make sense to me.

**Weaknesses:**

-  The contribution is incremental. The approach basically ensembles AnchorPoints, TinyBenchmarks, and GeneralScale. AnchorPoints already evaluated an embedding-based baseline (see "Pretrained" in Table 2). This paper builds on that idea by replacing pretrained embeddings with GPT-4o annotations (General Scales). Following TinyBenchmarks, it also uses a weighted average between two estimators.
- The proposed method does not necessarily solve the generalization problem. Previous approaches can indeed fail to transfer when behavior learned from one model family does not generalize to another. Similarly, the proposed approach relies on GPT-4o to annotate the 16 dimensions, and those annotations may also fail to generalize across different model families. The results in Table 2 and 3 also don't support the better generalization of the proposed method.
-  The IRT implementation differs from the original TinyBenchmarks paper. TinyBenchmarks Section 3.2 states that data points are clustered by the scores of existing LLMs. This paper, however, implements IRT by clustering IRT parameters, as stated in line 252.
-  The comparison is incomplete. For methods like AnchorPoints and IRT, one could reduce the number of training models instead of using 300 models; that comparison is missing. TinyBenchmarks introduces P-IRT and GP-IRT, both of which should be compared. The baseline MetaBench is mentioned but not compared.

**Questions:**

- Why does the IRT++ method become worse as the subset size increases?  This is inconsistent with the results in the TinyBenchmark paper, e.g., Figure 5.
- The Random baseline error looks unusually high. How was the Random baseline implemented? Since the OpenLLM leaderboard is multi-task, one should randomly sample data points from each task, compute the estimated score per task, and then average the scores across tasks. Otherwise, tasks with more data will dominate the sampled points, and the result will be misleading.

---

> ### Author Response · Authors · 2025-11-26
>
> Thank you for your review. We appreciate your concrete feedback and references to the relevant literature as relates to this paper. We have made substantial edits to the paper in response to your feedback and believe it has strengthened the result.
> At a high level the changes are:
>
> 	1. **New aggregation approach.** In response to your comments, rather than treating every item as equal, we are now weighting the items so that each subtask of the leaderboard is given equal weight. We have re-run all of the results with this change.
>
>         2. **Bugfix on the random baseline.** As you noticed, there had previously been a bug where the random baseline used too few points in the sample. The random baseline now uses the same number of points as the other methods and is stronger.
>
> Overall, we still find that our method achieves results comparable to the top baselines with far less upfront cost.
>
> Our work introduces a fundamental shift in efficient evaluation methodologies — moving from model-centric to item-centric subset selection. We believe this is not an incremental refinement; it represents a reconceptualization of how efficient benchmarking should be approached.
>
> While tinyBenchmarks established the problem formulation of efficient benchmarking, existing solutions—including tinyBenchmarks and anchor points—operate under a model-centric paradigm that selects items based on historical model performance. We are the first to identify the fundamental limitations of this paradigm, which requires a performance matrix $Y$ generated by running hundreds of models on the entire benchmark. This is the explicit source of: prohibitive upfront costs, cold-start failures on new benchmarks, and the fragile assumption that future models will replicate past failure patterns. More importantly, we are the first to propose a successful alternative item-centric paradigm where selection is based on the intrinsic properties of items themselves, independent of any model ensemble.
>
> This paradigm shift unlocks qualitatively different capabilities: immediate applicability to new benchmarks without previous model runs (cold start), invariance to across model architectures, and reduction in upfront selection costs. Within this new paradigm, we identify the General Scales methods as a valuable source of cognitive embeddings and demonstrate—for the first time—that item-centric approaches can match or exceed the predictive fidelity of model-centric methods while being dramatically more efficient.
> Regarding "pretrained embeddings" in AnchorPoints: The baseline you reference uses generic pretrained embeddings without cognitive grounding. We show that
>
> - Cognitively-grounded embeddings outperform generic ones
> - GNN distillation makes this practical at scale
>
> Regarding the weighted estimator: While we adopt the weighted averaging idea from prior work, we apply it to fundamentally different estimators. Our second estimator leverages the unique, interpretable nature of our cognitive scales by fitting per-dimension logistic regression (Lines 296-299), which is only possible with our item-centric approach.

---

> > ### Author Response · Authors · 2025-11-26
> >
> > **Regarding the tinyBenchmarks implementation:**
> > - Our description of the p-IRT and gp-IRT from tinyBenchmarks using the IRT parameters for clustering is correct as we described it. This is detailed in Section 4.1 of that paper. The portion that you refer to in Section 3.2 of that paper provides a generic description of anchor point-style methods and a criticism of using correctness as the embeddings.
> > - Our implementation is a fork from their public repo, and uses the same code to implement their methods. We have added repeated sampling due to the randomness of the initializations of the methods introduced in tinyBenchmarks, but have not changed how their method is implemented.
> > - We were also surprised to see IRT++ performing worse as the subset size increases (Question 1). Having changed the method of aggregation, this effect has gone away.
> >
> > **With regard to baselines:**
> > - The estimator that we call “IRT++” is the gp-IRT estimator, which was the top performing estimator from the tinyBench paper. For clarity, we have renamed it to gp-IRT. The gp-IRT estimator was described as “build[ing] upon p-IRT to overcome its limitations”, and as such should be strictly better. Since we already had multiple baselines, we prioritized showing the theoretical best one and avoiding noise. We have now added the p-IRT results in Appendix A. The results are generally poor, so we have not added them to the headline tables.
> >
> > - Regarding MetaBench, as noted in Footnote 1 , metaBench requires over 5,000+ model runs, making it an extreme example of the high-cost paradigm we are critiquing. We state, "due to the impracticality of having 5000 other model runs in most cases, we do not include their approach. If the upfront costs of MetaBench were not prohibitive it could be a relevant baseline, but we consider this to be unlikely to occur for all but the most popular public benchmarks. We provide this reference so that readers who are interested in these very popular benchmarks can find it, but the setting is not really equivalent in practice, and so we do not include it as a baseline.
> >
> > - In all cases, we have now **changed our method of aggregation** to treat each of the subtasks of the Open LLM Leaderboard as equal. We have re-run all of the results with this change and updated the paper.
> >
> > **With regard to generalization**, we would argue that our approach is inherently more flexible.
> > - Tables 3-5 show competitive performance with existing methods, and by focusing on an item-centric framing based on cognitive demands we break the fundamental assumption that previous models and their error patterns should be predictive of future models. The difference in performance between our model and the baselines is smaller here than in the headline results, suggesting a slight comparative advantage in this setting. We argue that task’s cognitive demands being stable across model families is a weaker and more interpretable assumption. Further improvement in item embeddings may be able to show even better generalization.
> > - Our annotations are model family agnostic and do not depend on GPT-4o specifically, but could be run with any annotator model, including our lightweight GNN. In particular, we explicitly address the dependency on a single annotator model (like GPT-4o) with our SCALES++ LITE variant. By distilling the annotation knowledge into a lightweight GNN , we create a fully open and non-proprietary method for generating cognitive embeddings, further enhancing robustness.
> >
> > Thank you again for your review.

---

> > > ### Comment · Reviewer_Gonq · 2025-11-27
> > >
> > > I appreciate the authors' efforts in addressing the rebuttal, although I think that re-running all the results during rebuttal already recommends resubmission. Here are my remaining concerns:
> > >
> > > - I do not view "reconceptualization" as a novel contribution. Methodologically, the claim that "cognitively grounded embeddings outperform generic ones" replaces the weak embeddings used in AnchorPoints baselines with a stronger embedding generated by ChatGPT. Therefore, I maintain my assessment of the contribution as 1.
> > >
> > > - Regarding the implementation of TinyBenchmarks, I agree with the authors after double-checking the corresponding code.
> > >
> > > - In the updated results, the proposed method cannot even consistently outperform Random Sampling. Additionally, the increasing estimation error as the subset size grows is also buggy to me, as shown in Figure 2 of the paper. Thus, I maintain my assessment of soundness as 1.

---

### Comment · Area_Chair_eD7J · 2025-11-27
**Reviewer Reminder: Author Rebuttals Available**

Dear Reviewers,

The authors have posted their rebuttals to your reviews.

Please read the authors' responses, assess whether your concerns have been addressed, and update your rating and confidence accordingly.

Your prompt attention to the rebuttals is appreciated.

Best,
AC

---

### Author Response · Authors · 2025-12-04
**General AC Response**

Dear AC,

Thank you for stepping in to cover the review process in the absence of further reviewer interaction. In this comment, we attempt to present a fair summary of the primary concerns of each of the reviewers in their initial reviews, as well as the changes that we have made to address each of them (indicated in bold). In order to keep this response short, we present both partial reviewer comments and excerpts from our responses, since the full comments and responses are available as well below.

Overall, we believe that our paper presents an important reconceptualization of the benchmark subset selection problem, focusing on an item-centric, rather than model-centric approach. This framing is largely novel, and is certainly the first implementation which shows competitive results with other existing approaches at 95% lower cost. In the review process, two reviewers also highlighted the significance of our approach. As far as the empirical results, we initially had a bug in the random baseline which we corrected, and we changed our aggregation approach to one favoured by one of the reviewers. With these changes, our method performs slightly worse than before, but is still competitive with the other established methods at lower cost. Our paper presents a valuable contribution to the efficient benchmarking literature, opening up a new direction of exploration in item-centric methods.

---

> ### Author Response · Authors · 2025-12-04
> **Reviewer 1 (Gonq)**
>
> This reviewer was the most pessimistic about our work, and felt that the contribution was “too incremental”. They also raised specific points about the baselines and implementation that they did not like. They did believe that our paper “highlighted important issues with evaluations” and presented our work clearly.
>
> *To address the comments of this reviewer, we:*
> - Changed the aggregation methods used in our scoring to one which they preferred and re-ran the results
> - Added additional baselines
>
> **On the specific weaknesses and questions:**
>
>      The contribution is incremental.
>
> Our work introduces a fundamental shift in efficient evaluation methodologies — moving from model-centric to item-centric subset selection. We believe this is not an incremental refinement; it represents a reconceptualization of how efficient benchmarking should be approached.
>
> Existing solutions—including tinyBenchmarks and anchor points—operate under a model-centric paradigm that selects items based on historical model performance. We are the first to identify the fundamental limitations of this paradigm, which requires a performance matrix $Y$ generated by running hundreds of models on the entire benchmark. This is the explicit source of: prohibitive upfront costs, cold-start failures on new benchmarks, and the fragile assumption that future models will replicate past failure patterns. We are the first to propose a successful alternative item-centric paradigm where selection is based on the intrinsic properties of items themselves, independent of any model ensemble.
>
>     The proposed method does not necessarily solve the generalization problem.
>
> We would argue that our approach is inherently more flexible. Tables 3-5 show competitive performance with existing methods. The difference in performance between our model and the baselines is smaller here than in the headline results, suggesting a slight relative advantage in this setting. We argue that task’s cognitive demands being stable across model families is a weaker and more interpretable assumption. Further improvement in item embeddings may be able to show even better generalization.
>
>     The IRT implementation differs from the original TinyBenchmarks paper. TinyBenchmarks Section 3.2 states that data points are clustered by the scores of existing LLMs. This paper, however, implements IRT by clustering IRT parameters, as stated in line 252.
>
> *This is a factual error by the reviewer.* Our description of the p-IRT and gp-IRT from tinyBenchmarks using the IRT parameters for clustering is correct as we described it. This is detailed in Section 4.1 of that paper. The portion that they refer to in Section 3.2 of that paper provides a generic description of anchor point-style methods and a criticism of using correctness as the embeddings.
>
>     The comparison is incomplete. For methods like AnchorPoints and IRT, one could reduce the number of training models instead of using 300 models; that comparison is missing. TinyBenchmarks introduces P-IRT and GP-IRT, both of which should be compared. The baseline MetaBench is mentioned but not compared.
>
> The estimator that we call “IRT++” is the gp-IRT estimator, which was the top performing estimator from the tinyBench paper. **For clarity, we have renamed it to gp-IRT.** We had run p-IRT, but the results are generally poor, so we had not added them to the headline tables. **We have now added them in Appendix A.** The metaBench paper points out that gp-IRT is already training models with far more parameters than data - training gp-IRT on 16 examples would raise serious questions about the quality of the fit. We have omitted metaBench from our baselines because it requires 5000 full model runs to train. If the upfront costs of MetaBench were not prohibitive it could be a relevant baseline, but we consider this to be unlikely to occur for all but the most popular public benchmarks.
>
>     Why does the IRT++ method become worse as the subset size increases? This is inconsistent with the results in the TinyBenchmark paper, e.g., Figure 5. AND The Random baseline error looks unusually high. Since the OpenLLM leaderboard is multi-task, one should randomly sample data points from each task, compute the estimated score per task, and then average the scores across tasks. Otherwise, tasks with more data will dominate the sampled points, and the result will be misleading.
>
> We found that this arises due to a change in the aggregation methods. Our original submission used a straight average across all benchmark items in the Open LLM Leaderboard. The previous tinyBenchmark paper used a weighted average so that each subtask received equal weight. **We have changed our aggregation method to the weighted average used in tinyBenchmarks.** This change was bad for the performance of our Scales++ method relative to the tinyBenchmarks baseline, but the overall argument that our method is of comparable accuracy with less upfront cost remains true.

---

> ### Author Response · Authors · 2025-12-04
> **Reviewer 2 (pzPu)**
>
> This reviewer was the most positive about our paper, and gave it good scores for soundness, presentation and contribution. They particularly liked the “innovative” example-centric approach, and thought we introduced and compared to previous works well. They did raise clarification questions about what we did.
>
> *To address the comments of this reviewer we edited the text to clarify the answers to their questions.*
>
> **On the specific weaknesses and questions:**
>
>     Can you elaborate more on some design choices you made?
>
> We provided a longer response below and **edited the text to provide more details.**
>
>     Can you elaborate more on the auxiliary training data you used?
>
> We used 8000 samples from TULU-SFT. **We re-wrote this section to make it clearer.**
>
>     Can you explain in figure 2, what are included in compute cost for each method? Is "initial LLM calls" equal to 16 for scales++ and 300 for IRT methods?
>
> We provided a longer explanation below. The reviewer interpreted the figure correctly.
>
>     I'm quite confused at section 5 where some cross-architecture and cross-model-size experiments were done. In my understanding scales does not require any training on "source models", so what does "cross-architecture" refer to here?
>
> The reviewer is correct, our method does not require source models. **We edited the text to make the experiments easier to follow, and explicitly noted that the source models only matter for the baselines.**

---

> ### Author Response · Authors · 2025-12-04
> **Reviewer 3 (4Sk9)**
>
> This reviewer felt that the contribution of our paper was valuable, with “strong” empirical results. They raised issues with the presentation, and a bug in the random baseline.
>
> *To address the comments of this reviewer we:*
> - Fixed the bug in the random baseline
> - Revised and copy-edited the text of the paper
> - Added a clearer limitations section
>
> **On the specific weaknesses and questions:**
>
>     It is quite difficult to understand the experimental setup for most of the experiments in section 5.
>
> **We have re-written these sections to make them clearer.**
>
>     I suspect that the evaluation of the random baseline in figure 2 is bugged.
>
> The reviewer gave helpful comments on locating the bug, and we found and fixed it. (The baseline was not correctly increasing the number of random points being sampled.) **We fixed the bug and re-ran the baseline.**
>
>     There is no meaningful discussion of limitations. Given that the method does not come with any guarantees (and seems to severely underperform alternatives in some settings, according to the appendix), disclaimers about how to responsibly use this should be included in the main text.
>
> **We added a limitations section,** discussing the performance of the model under different conditions. **We also moved the smaller benchmarks into the main text** to ensure that the reader is given an accurate understanding of the strengths and weaknesses of our approach.
>
>     Beyond that, the paper would strongly benefit from a thorough proofread.
>
> **We went through and more carefully edited the paper.**
>
>     What is the setting with 16 initial LLM calls for IRT reported in Table 2? Also, why are these results not reported in Figure 2 (IRT seems to outperform the proposed method for >0.5% data points here)?
>
> Table 2 shows an experiment with transfer between two different test cases. The 16 example result is from training only on MoE models to predict the Dense ones. Since we have no principled reason to select this subset for training, we attribute these results to chance and do not treat them as an additional baseline.

---

### Meta-Review · Area_Chair_jXuf · 2025-12-16

**Summary:**

This paper addresses the high cost of comprehensive LLM evaluation by proposing an item-centric method for constructing tiny yet representative benchmarks. Departing from model-centric selection, the proposed method Scales++ selects benchmark items based on their intrinsic cognitive demands, enabling efficient, interpretable, and cold-start–friendly evaluation. Experiments confirm the effectiveness of the proposed method in several cases.

This paper received comments from three reviewers. After the rebuttal, two reviewers who were previously negative remain negative reviewers. A reviewer who was previously positive has also raised some new concerns. The technical contribution of this work is somewhat weak. Besides, experiments in the current form are not convincing, as reflected in both experimental settings and results.

Given these issues, the AC recommends rejection. The authors are encouraged to address the mentioned concerns to enhance the clarity and overall impact of the work in future submissions.

**Reviewer Concerns:**

- For Reviewer Gonq, the concerns of method implementation are partially resolved. The concerns in the work's contributions and experimental results are still outstanding.
- For Reviewer pZpu, the questions of the method details and experiments are solved. However, due to the significant change in experimental results, new concerns about the results are raised.
- For Reviewer 4sK9, the concerns about the experimental setup and paper presentation are partially resolved. However, the experimental results in the current form are still not convincing.

**Reviewer Scores:**

- **Reviewer Gonq.**  The main concerns of this reviewer were not addressed well. He/she is still negative about the current form. Therefore, the score would not be changed, even if the reviewer participates fully in the discussion.
- **Reviewer pZpu.** The experimental results and significant changes in this work make readers confused and worried about the contribution of this work. Although this reviewer was positive initially, he/she may lower the score due to the mentioned issues.
- **Reviewer 4sK9.** After the rebuttal, the concerns about the empirical evaluation are still outstanding. Therefore, this reviewer would not change the score with a high probability.

---

### Decision · Program_Chairs · 2026-01-26

Reject